# Investigation of an Efficient Multi-Class Cotton Leaf Disease Detection Algorithm That Leverages YOLOv11

**DOI:** 10.3390/s25144432

**Published:** 2025-07-16

**Authors:** Fangyu Hu, Mairheba Abula, Di Wang, Xuan Li, Ning Yan, Qu Xie, Xuedong Zhang

**Affiliations:** 1College of Information Engineering, Tarim University, Alaer 843300, China; 15945632454@163.com (F.H.); 18095809057@163.com (M.A.); 17590696276@163.com (D.W.); 17860530361@163.com (X.L.); 18295549487@163.com (N.Y.); 2Key Laboratory of Tarim Oasis Sensors, Ministry of Education, Tarim University, Alaer 843300, China

**Keywords:** YOLOv11, U-Net v2, cotton leaf disease detection

## Abstract

Cotton leaf diseases can lead to substantial yield losses and economic burdens. Traditional detection methods are challenged by low accuracy and high labor costs. This research presents the ACURS-YOLO network, an advanced cotton leaf disease detection architecture developed on the foundation of YOLOv11. By integrating a medical image segmentation model, it effectively tackles challenges including complex background interference, the missed detection of small targets, and restricted generalization ability. Specifically, the U-Net v2 module is embedded in the backbone network to boost the multi-scale feature extraction performance in YOLOv11. Meanwhile, the CBAM attention mechanism is integrated to emphasize critical disease-related features. To lower the computational complexity, the SPPF module is substituted with SimSPPF. The C3k2_RCM module is appended for long–range context modeling, and the ARelu activation function is employed to alleviate the vanishing gradient problem. A database comprising 3000 images covering six types of cotton leaf diseases was constructed, and data augmentation techniques were applied. The experimental results show that ACURS-YOLO attains impressive performance indicators, encompassing a mAP_0.5 value of 94.6%, a mAP_0.5:0.95 value of 83.4%, 95.5% accuracy, 89.3% recall, an F1 score of 92.3%, and a frame rate of 148 frames per second. It outperforms YOLOv11 and other conventional models with regard to both detection precision and overall functionality. Ablation tests additionally validate the efficacy of each component, affirming the framework’s advantage in addressing complex detection environments. This framework provides an efficient solution for the automated monitoring of cotton leaf diseases, advancing the development of smart sensors through improved detection accuracy and practical applicability.

## 1. Introduction

Cotton is a critically important plant fiber crop both in China and on a global scale. It is cultivated in more than 50 countries, covering an extensive geographical area. Among the world’s top five cotton-producing countries, China holds the leading position in terms of production volume [1]. According to the data collected from 2017 to 2019, the loss rate of cotton production attributed to pest infestations and adverse environmental conditions was substantial [2]. In the process of cotton cultivation, approximately 90% of disease symptoms are observed on cotton leaves, encompassing conditions such as cotton boll blight, cotton anthracnose, and cotton leaf mildew. These diseases impair plant physiological functions and disrupt the vascular system, thereby directly leading to reduced cotton yields and degraded fiber quality. In this context, the adoption of automated detection technology for large–scale cotton crop monitoring carries considerable importance in reducing disease–induced losses and facilitating the sustainable growth of the cotton sector [3].

In traditional cotton leaf disease detection methods, diagnosing cotton diseases mainly involves analyzing the characteristic features of cotton leaves [4]. However, the identification of leaf diseases with the human eye is prone to errors due to the similarity between some diseases, making it challenging for humans to identify diseases accurately. This method also requires significant labor input and exhibits strong subjectivity, which renders it unsuitable for meeting the large-scale agricultural demands of today. In contrast, automated disease detection has undergone significant evolution, progressing from early machine learning techniques to contemporary deep learning approaches. Machine learning achieved notable breakthroughs in crop disease classification during its early stages. Its main strength resides in the markedly improved precision and effectiveness in plant pathological classification relative to previous approaches, thereby broadening its scope of applicable scenarios [5]. Traditional machine learning shows limited generalization for diverse disease symptoms [6]. With the advancement of intelligent sensors in China, the replacement of human labor with machines has emerged as an inevitable trend. The integration of state-of-the-art tech into sensors, including AI, 5G, human–machine interaction, and big data, promotes the use of computer vision and deep learning in agricultural advancement [7].

The integration of deep learning technology into the agricultural sector is progressively advancing, particularly in the areas of crop disease detection and classification. Deep learning models, particularly convolutional neural networks (CNNs), have been widely utilized in sensors for the detection and categorization of pests and diseases that are harmful to specific crops [8]. These models are capable of processing and analyzing vast amounts of image data, automatically extracting relevant features, and performing precise disease classification, which is critical for the early detection and prevention of crop diseases. Through the analysis of satellite imagery and drone-captured field images, CNNs can autonomously identify crop species, assess their health and growth status, and monitor vegetation indices, such as the Normalized Difference Vegetation Index (NDVI), to evaluate crop conditions [9]. However, crops may simultaneously be affected by multiple pests and diseases during various growth stages. The identification of individual pests or diseases is no longer sufficient to meet the needs of growers. Consequently, researchers have begun to focus on combinations of pests and diseases that more frequently co-occur within the same growth stage. By employing enhanced deep learning models, they extract features of these pests and diseases and classify them accordingly.

These studies have enabled growers to utilize a single model for distinguishing among multiple plant pests and diseases, thereby promoting advancements in agricultural applications. Within the domain of sensors, deep learning technology has exhibited considerable potential, with particular emphasis on the detection and classification of crop diseases [10]. As crops may encounter the challenge of multiple pests and diseases at various growth stages, researchers are utilizing advanced deep learning models to identify and classify these diseases, thereby addressing growers’ requirements for multi-disease identification. In scenarios with limited datasets, the model’s generalization capability and recognition accuracy are enhanced by leveraging public plant datasets and transfer learning technologies [11]. The advancement of these technologies not only enhances the precision of crop monitoring but also refines crop management strategies, facilitates the development of precision sensors, and strengthens quality control of agricultural products. As technology continues to evolve and datasets become increasingly diverse, the application of deep learning in sensors is expected to expand further and delve deeper into various domains.

Deep learning-based detection models, such as the YOLO series, have demonstrated superior performance in agricultural pest and disease monitoring, particularly in detecting leaf diseases in crops like cotton [12]. These models are capable not only of identifying the disease type but also of precisely locating the disease on the leaf, which is critical for the targeted application of treatments and the safeguarding of crop health [13]. The high adaptability of these models in scenarios involving multiple diseases allows them to maintain consistent recognition performance across varying environmental conditions, such as fluctuating illumination and complex backgrounds, thus providing farmers with a dependable monitoring solution.

To further enhance the practicality of these models, researchers are investigating methods to enable their functionality under more complex real-world field conditions. This has been accomplished by integrating leaf images of pests and diseases—captured with various natural backgrounds—into the training dataset, which in turn enhances the model’s resilience in complex environments [14]. This approach enhances the model’s ability to accurately identify and locate diseases in practical scenarios, even when leaves are partially occluded or environmental conditions are suboptimal.

To satisfy the real-time and convenience demands of agricultural field operations, researchers are actively exploring methods to optimize the model, aiming to minimize computational resource consumption while maintaining high accuracy and accelerating detection speed. This involves designing more lightweight model architectures and refining algorithms for compatibility with mobile and embedded devices. Such advancements not only improve the efficiency of pesticide application and mitigate its environmental impact but also enhance the overall yield and quality of crops [15]. The integration of deep learning technology into agricultural pest detection is advancing steadily, enabling farmers to achieve more precise crop protection and management through rapid and accurate disease identification and localization. As technology continues to evolve, these models are anticipated to play an increasingly significant role in enhancing the efficiency and sustainability of agricultural production [16].

In the domain of modern agricultural disease management, YOLO series models have emerged as a pivotal technology for outdoor real-time monitoring due to their high accuracy and substantial advantages in single-stage target detection [17]. These models demonstrate a remarkable capacity to rapidly adapt to the instabilities inherent in outdoor environments, such as fluctuations in lighting and background disturbances. This guarantees consistent and dependable detection results under diverse conditions. Ongoing progress in the YOLO series, which encompasses components such as PGI and GELAN in YOLOv9, along with the C3k2 module and C2PSA module in YOLOv10 and YOLOv11, has significantly boosted the model’s capacity for addressing complex detection tasks [18]. Notably, these improvements have been particularly effective in detecting small and occluded objects. Furthermore, the incorporation of attention mechanisms, such as the SE module, enables the model to prioritize critical image features, thereby improving the accuracy of disease recognition under challenging field conditions. The integration of these technologies not only elevates the detection performance of the model but also enhances its robustness in dynamic environments [19]. These developments suggest that YOLO series models will play an increasingly significant role in smart sensors, contributing to the improvement of crop management efficiency and effectiveness while fostering the sustainable development of sensors.

While YOLO series models are widely employed in the object detection domain owing to their fast detection speed and strong real-time capability, some constraints still exist when they are applied in cotton leaf disease detection [20]. First, the YOLO series models exhibit constrained performance in small target detection. Given that some cotton leaf diseases manifest as small lesions during their initial stages, this often leads to missed detections or false positives. Second, the model’s target detection capability in the presence of complex backgrounds requires enhancement. Cotton plants possess diverse leaf shapes, and the growth environment includes interference factors such as weeds, which complicates the accurate identification of cotton leaf diseases. Additionally, the YOLO series models demonstrate insufficient generalization ability regarding disease features [21]. Cotton leaf diseases vary significantly across different growth stages and environments, further reducing the model’s detection accuracy in new scenarios or with novel disease types. Consequently, these challenges hinder the models from meeting the stringent requirements for precise cotton leaf disease detection.

To address the abovementioned issues, considering the constraints of YOLO series models in cotton leaf disease detection and the complexity of such detection scenarios, there is an urgent demand for innovative advances in research. The U-Net v2 model, which is well known in the field of medical image segmentation for its unique encoder–decoder structure and strong feature extraction capacities, has offered new perspectives [22]. Drawing inspiration from these findings, the current study introduces the U-Net v2 module into the domain of cotton leaf disease detection. Leveraging the advanced YOLOv11 as the base architecture, the framework was refined through multiple approaches: integrating an attention mechanism to guide the model in focusing on critical disease features; comprehensively optimizing modules such as SPPF and C3k2 to enhance the network’s capacity for capturing disease-related information across diverse scales; and improving the activation function to strengthen the model’s nonlinear expression capability. Through a series of innovations and integrations, the ACURS-YOLO network was developed to address contemporary requirements, aiming to resolve the challenge of cotton leaf disease detection via novel architectural design and algorithmic optimization. The primary contributions of this research are outlined as follows:A dataset comprising 3000 images of six typical cotton leaf diseases was built, and the effectiveness of the improved model was verified via data augmentation methods within a scientifically structured experimental setup.To tackle the challenges of complicated background disturbances, failure to identify small target lesions, and inadequate adaptability to diverse disease types in cotton leaf disease identification, the U-Net v2 module from the medical image segmentation field was incorporated into the YOLOv11 core network. Furthermore, this SimSPPF component was engineered to replace the conventional SPPF, thus lowering computational load while maintaining multi-scale feature extraction capacity and boosting inference speed. This C3k2_RCM component was embedded in the neck network by integrating a rectangular self-calibration mechanism to strengthen long-range contextual modeling. Furthermore, the ARelu activation function was employed to alleviate gradient vanishing problems, thereby achieving simultaneous improvements in detection accuracy and training stability.The ACURS-YOLO network is benchmarked against the YOLO series to evaluate and validate the disease detection capability and overall performance of the proposed model in cotton leaf disease scenarios.

The remainder of this paper is structured as follows: Section 2 details the materials and methods, including dataset construction, the model architecture, and the experimental setup. Section 3 presents the experimental results, including performance comparisons with classical models, ablation studies, and model testing in complex scenarios. Section 4 discusses the implications of the results, limitations of the current study, and future research directions. Finally, Section 5 summarizes the key findings and contributions of this work.

## 2. Materials and Methods

### 2.1. Experimental Design

Our experimental setup adopts a research framework commonly employed in computer vision tasks, with the implementation process shown in Figure 1. This serves to empirically verify the validity of the upgraded model for cotton leaf disease detection. In this study, a self-constructed dataset was developed to encompass six representative cotton leaf diseases (Cotton Leaf Spot, Cotton Leaf Curl, Cotton Brown Spot, Cotton White Mold, Cotton Verticillium Wilt, and Cotton Fusarium Wilt). The original images were acquired by integrating field photographs with publicly available datasets. To ensure data quality, the original images underwent preprocessing steps, including cleaning and labeling, to generate disease identification labels. By associating each image with its corresponding label, a specialized dataset for cotton leaf disease detection was established. Following dataset partitioning, the ACURS-YOLO network was constructed, and the model was utilized to train, validate, and test the dataset. A comprehensive set of evaluation metrics was employed to compare the enhanced model against several baseline models and mainstream architectures. Additionally, ablation studies were conducted to assess the contribution of each improved module. Ultimately, a radar chart was utilized to visually represent the performance characteristics of the model, thereby ensuring the scientific rigor and interpretability of the model optimization strategy.

### 2.2. Dataset Construction

The data source for this study comprises publicly available image datasets of cotton leaf diseases from the Internet and a cotton plantation base located in Alar City, Xinjiang Province, China. This dataset includes six common types of cotton leaf diseases: Cotton Leaf Spot, Cotton Leaf Curl, Cotton Brown Spot, Cotton White Mold, Cotton Verticillium Wilt, and Cotton Fusarium Wilt. A total of 1200 high-quality images under various environmental conditions have been selected for analysis, as illustrated in Figure 2. Since raw RGB images are not directly usable for training, validation, or testing in object detection tasks, a combination of source images and their matching reference labels must be employed. The labels indicate the locations of the objects to be detected within the images. Therefore, prior to training, validation, or testing, the original data must undergo labeling to associate each image with its respective annotations [23].

In this study, LabelImg was utilized to annotate the original images in YOLO format. The annotation files were generated in XML format, which included details such as image dimensions, label names, and location information. For instance, the labeling of Cotton Leaf Spot is presented in Figure 3. To simplify the fitting process, data collection was randomly divided into a training group, a validation group, and a test group in an 8:1:1 proportion. Specifically, the training group was employed for model fitting, the validation group served to fine-tune hyperparameters during the fitting phase, and the test group was used for the final model evaluation [24].

To boost the model’s adaptability in diverse settings, enhance its generalization performance, and alleviate the negative impacts of environmental factors, data augmentation methods were applied to the dataset. Such methods involved image translation, adding noise, flipping, local brightness adjustment, and mirroring. As a result, the experimental dataset consisted of 3000 cotton leaf images covering six distinct diseases, with details provided in Table 1.

### 2.3. Methods

#### 2.3.1. YOLOv11 Image Detection Model

Since the primary objective of this experiment is to achieve leaf disease detection, a target detection method was employed. Classical region-based object detection algorithms, such as Faster R-CNN [25], the SSD object detection algorithm [26], and YOLO versions (e.g., YOLOv8, YOLOv9, and YOLOv10), are capable of detecting objects. In this experiment, the YOLOv11 model was selected for cotton leaf disease detection. As depicted in Figure 4 [27], YOLOv11 comprises four core parts, including the input, backbone, neck, and head, which serve the following functions [28]:During the input stage of YOLOv11, data preprocessing tasks are performed, including resizing, normalizing, and adapting input images to ensure that the model can effectively process images of varying sizes and formats. This preprocessing step is essential for enabling YOLOv11 to extract features more efficiently during training and significantly contributes to the model’s generalizability and stability.Backbone: The backbone network in YOLOv11, which is responsible for feature extraction and delivery, primarily consists of the Conv, C3K2, SPPF, and C2PSA modules. The Conv module retains a structure similar to that of previous generations in YOLOv11 and performs operations such as image downsampling, reducing spatial dimensions, increasing the number of channels, and processing input and output data. The C3K2 module integrates the functionalities of the C2f and C3 modules, leveraging the C2f structure to achieve faster data processing speeds. SPPF represents an enhanced version of spatial pyramid pooling, maintaining the performance of traditional SPP while significantly reducing computational complexity and accelerating calculations. C2PSA is a dedicated module that is engineered to strengthen the attention mechanism and boost feature extraction capabilities, thereby elevating the model’s detection precision for objects of different sizes [29].Neck: The neck network in YOLOv11 serves to fuse multi-scale features transmitted from the backbone. By incorporating advanced techniques, such as the attention mechanism and bidirectional feature pyramid networks, it enhances both the detection performance and computational efficiency of the model.Head: The head processes the feature map transmitted by the neck to facilitate the classification and recognition of training outcomes.

#### 2.3.2. ACURS-YOLO Object Detection Model

Because the cotton planting environment is highly complex, factors such as mutual occlusion between cotton leaves, similar textures, and overlapping areas significantly impact the accuracy of model-based cotton leaf disease detection. Drawing inspiration from medical image segmentation techniques, we propose an ACURS-YOLO network specifically designed for cotton leaf disease detection. This model enhances the backbone network of YOLOv11 by integrating the UNetv2 architecture, which is widely used in medical imaging. Furthermore, it utilizes the CBAM attention mechanism to polish segmented features, which in turn enhances the model’s comprehensive detection performance for cotton leaf diseases. The conventional SPPF component has been replaced by SimSPPF (Simplified SPPF), alongside which a rectangular self-calibration component is inserted following the C3k2 component. Furthermore, the ARelu activation function has been introduced to optimize performance. ACURS-YOLO’s structural framework is illustrated in Figure 5. It consists of four core parts: the input, backbone, neck, and head. After establishing the experimental environment, this dataset is preprocessed through its input module. Subsequently, feature extraction and feature fusion are performed through the backbone and neck modules. Finally, the head module enables the classification of cotton leaf diseases.

#### 2.3.3. U-Net v2

In recent years, there have been swift advances in the area of image segmentation within computer vision, with notable progress, especially in the medical field. U-Net demonstrates high practicality for medical image segmentation, serving as a primary tool for such tasks [30]. Enhanced the U-Net architecture to address the limitation of its connection method in effectively integrating low-level and high-level features. Simultaneously, they adopted a more sophisticated approach to refine high-level features, thereby enhancing the extraction of key region features and improving model accuracy. By incorporating this module into the backbone section of YOLOv11, the successful application experience of U-Net in the medical field was effectively adapted for detecting cotton leaf diseases. The structural diagram of the U-Net v2 module is presented in Figure 6 [31].

The following is an introduction to the internal modules of U-Net v2:

Encoder: This is used to receive image I from the input module and collect feature maps in the SDI.

SDI and Decoder: Each hierarchical feature map generated by the encoder is applied to the CBAM. The formula is as follows:(1)$$f_{i}^{1}=(\phi_{i}^{c}\circ \varphi_{i}^{s})(f_{i}^{0})$$

$f_{i}^{1}$ denotes the feature map processed by the ith layer, where $\Phi_{i}^{c}$ signifies the spatial attention mechanism parameters of the ith layer, and $\varphi_{i}^{s}$ indicates the channel attention mechanism parameters of the ith layer. Subsequently, the reference target is transmitted to the decoder to align with the corresponding resolution, as described below:(2)$$f_{ij}^{3}=\left\{\begin{array}{c} A(f_{j}^{2},(H_{i},W_{i}))\;\;\;\;\;\;\;\;if\;j<i,\\ B(f_{j}^{2})\;\;\;\;\;\;\;\;\;\;\;\;\;\;\;\;\;\;\;\;\;\;\;\;\;\;if\;j=i,\\ C(f_{j}^{2},(H_{i},W_{i}))\;\;\;\;\;\;\;\;if\;j>i, \end{array}\right.$$

In this formulation, *A*, *B*, and *C* represent the resolution scales associated with adaptive mean pooling, identity projection, and the bilinear resampling of $f_{j}^{2}$ to $H_{i}$ and $W_{i}$, respectively, where *i* ≥ 1 and *j* ≤ M. Subsequently, $f_{ij}^{3}$ undergoes modification via a 3 × 3 convolution, as specified below:(3)$$f_{ij}^{4}=\theta_{ij}\circ f_{ij}^{3}$$

$f_{ij}^{4}$ represents the j-th smoothed feature map of the i-th layer, and $\theta_{ij}$ represents the smoothed convolution parameters. To enhance the features at level *i* with richer semantic information and more precise details, the Hadamard product is applied to all feature maps adjusted to the same resolution. This process is formulated as follows:(4)$$f_{i}^{5}=H([f_{i1}^{4},f_{i2}^{4},f_{i3}^{4},\cdot \cdot \cdot ,f_{iM}^{4}])$$
where *H()* represents the Hadamard product, which is passed to the ith-level decoder for further resolution reconstruction and segmentation.

The selection of operations *A* (adaptive average pooling), *B* (identity projection), and *C* (bilinear resampling) is tailored to the multi-scale nature of cotton leaf lesions, which range from small early-stage spots to large mature lesions. Adaptive pooling (*A*) handles downsampling for high-level semantic alignment, identity projection (*B*) preserves the fine-grained details of medium-sized lesions, and bilinear resampling (*C*) upscales low-resolution features to match lesion edges in complex backgrounds, thereby ensuring consistent feature fusion across scales.

#### 2.3.4. CBAM

In the detection of cotton leaf diseases, the high similarity among various leaf diseases and the complexity of the environment pose significant challenges to the accuracy of target detection. Introducing the attention mechanism into the backbone of YOLOv11 represents an effective approach to leveraging the attention mechanism to extract features associated with cotton leaf diseases, thereby enhancing the precision of disease detection. The Convolutional Block Attention Module (CBAM), which integrates a channel-wise attention component and a spatial attention component, serves as a classic example of this method, as depicted in Figure 7 [32].

The channel-wise attention component works to reinforce feature expression in the channel dimension, while the spatial attention component acts to boost this kind of expression in the spatial dimension.

The functioning of the channel-wise attention component mainly entails performing overall maximum pooling and mean pooling [33]. The extracted features are subsequently processed through a Shared Multi-Layer Perceptron (MLP) to generate the feature vector. This vector’s formulation is expressed as follows:(5)$$M_{c}(F)=\sigma (MLP(AvgPool(F))+MLP(MaxPool(F))$$
where σ denotes the sigmoid function [34]. $M_{c}(F)$ represents the channel attention weight map, highlighting disease-related channels.

The aim of the spatial attention module is to emphasize the importance of different locations within an image. To start with, the features derived from feature maps are merged along the channel axis after being subjected to both max pooling and average pooling. Following this step, spatial attention weights are produced via convolutional layer handling. Consequently, this process enables the identification of salient image regions. The expression is as follows:(6)$$M_{s}(F)=\sigma (f^{7\times 7}([AugPool(F);MaxPool(F)]))$$

$f^{7\times 7}$ means that the size of the convolution kernel is *7 × 7*. $M_{s}(F)$ represents the spatial attention weight map, which emphasizes the lesion area in the spatial dimension.

The *7 × 7* convolution kernel in the spatial attention module is chosen to accommodate the typical spatial distribution of cotton leaf diseases: lesions often spread within a 7–10 pixel radius from their center, and the larger kernel captures contextual cues (e.g., leaf veins adjacent to lesions) in cluttered backgrounds, thereby reducing interference from irrelevant textures.

#### 2.3.5. SimSPPF

In the YOLOv11 architecture, the Spatial Pyramid Pooling—Fast (SPPF) module enhances the model’s adaptability to variations in target size via multi-scale pooling operations [29,35,36]. Its primary function is to extract features from different receptive fields using serial max pooling layers (e.g., a 5 × 5 pooling kernel) and to integrate multi-scale information through feature concatenation. This module is situated at the end of the backbone network and plays a critical role in aggregating global context information, thereby improving the detection performance for small objects, such as minute disease spots on cotton leaves. Despite its effectiveness in feature fusion, the SPPF module has certain limitations. The serial stacking of pooling layers introduces additional computational overhead, which is particularly pronounced when processing high-resolution images, such as those with dimensions of 2024 × 1536 pixels depicting field leaf images. This limitation adversely affects inference speed. SPPF employs the Sigmoid Linear Unit (SiLU) activation function, which introduces exponential operations and thereby increases computational complexity. Additionally, serial pooling results in strong dependencies between feature maps, limiting the potential for hardware-based parallel acceleration. To address these issues, the improved SPPF module simplifies its architecture, optimizes the activation function, and significantly enhances computational efficiency while preserving multi-scale feature extraction capabilities. In the backbone section, we replace the conventional SPPF module with SimSPPF (Simplified SPPF), as illustrated in Figure 8 [37].

The SimSPPF module is designed to streamline the computational process while preserving or even enhancing the capability to extract multi-scale features. In contrast, the traditional SPPF module processes input feature maps through multiple max pooling layers of varying sizes and subsequently concatenates these pooled feature maps along the channel dimension. While this approach effectively captures feature information across different scales, it also leads to an increase in computational load and model complexity. The SimSPPF module refines the design of the traditional SPPF by adopting a more streamlined architecture. This reduces computational complexity by decreasing the number of pooling layers and optimizing the pooling operation combinations. Furthermore, the SimSPPF module places greater stress on preserving and boosting feature information in the course of pooling, thus enhancing the model’s detection capacity for cotton patch disease recognition in various situations.

Serial pooling is replaced with parallel pooling, and multi-scale feature extraction is achieved via three parallel 5 × 5 max pooling layers. This design mitigates computational dependency and enhances GPU parallel efficiency. The formula is presented as follows:(7)$$F_{out}=Concat\left({MaxPool}_{5}^{\left(1\right)}\left(F_{in}\right),{MaxPool}_{5}^{\left(2\right)}\left(F_{in}\right),{MaxPool}_{5}^{\left(3\right)}\left(F_{in}\right)\right)$$

${MaxPool}_{5}^{\left(1\right)},{MaxPool}_{5}^{\left(2\right)},\;and\;{MaxPool}_{5}^{\left(3\right)}$ represent three parallel 5 × 5 max pooling operations designed to extract the multi-scale features of small lesions. $F_{out}$ and $F_{in}$ indicate the input and output feature maps, respectively.

The 5 × 5 max pooling kernel is selected based on a statistical analysis of the dataset: over 85% of small lesions (diameter < 15 pixels) fit within a 5 × 5 receptive field, balancing the need to capture lesion details with computational efficiency. Parallel pooling ensures multi-scale feature extraction without sacrificing speed, which is critical for the real-time detection of small, scattered lesions.

Activation function replacement: The SiLU activation function is replaced with ReLU, thereby eliminating the need for exponential operations and enhancing the convergence speed of training. The formula is presented as follows:(8)$$f_{ReLU}\left(x\right)=max\left(0,x\right)$$

#### 2.3.6. C3k2_RCM

Within object detection algorithms, the design of the neck module holds vital significance for multi-scale feature fusion and contextual information modeling. The original neck component C3k2 in YOLOv11 is based on the Cross-Stage Partial Connection (CSP) architecture, which achieves efficient feature fusion via lightweight residual connections and feature concatenation. However, it exhibits limitations in long-range contextual modeling and complex scene feature representation. To tackle this problem, this study introduces a neck architecture that combines efficient feature fusion and adaptive context enhancement by inserting a Rectangular Self-Calibration Module (RCM) following the C3k2 component. This approach significantly improves model performance for multi-scale targets, particularly in cotton leaf disease recognition. The original C3k2 module utilizes a convolution layer that features a kernel size of 2 to balance spatial resolution and computational expense. Its core architecture comprises a downsampling layer and multiple sets of bottleneck blocks. The input features are split into two branches: one branch extracts high-level semantics via lightweight residual transformations, while the other directly performs cross-stage concatenation to preserve low-level spatial details. Feature fusion is subsequently achieved through channel concatenation. This design mitigates the issue of gradient vanishing while ensuring computational efficiency. However, the feature interaction method based on local convolution and fixed-scale residual connections struggles to capture long-range dependencies between different levels of features, leading to insufficient modeling of semantic associations between targets and backgrounds in complex scenes. Additionally, the original model exhibits a limited capacity for adaptively adjusting the importance of features across channels, which may result in the weakening of key semantic features or the introduction of redundant information. This issue is particularly pronounced in small object detection, where the sparsity of contextual cues in low-resolution feature maps becomes more significant. To overcome these constraints, this study presents the C3k2_RCM module, which combines the RCM module with the C3k2 module to facilitate the in-depth extraction of multi-scale contextual information via rectangular self-calibration attention, shape self-calibration capabilities, and a feature fusion mechanism [38,39]. The architecture of the C3k2_RCM module is depicted in Figure 9.

#### 2.3.7. ARelu

The activation function plays a crucial role in various aspects of the performance of object detection models. By default, the SILU activation function is employed in YOLOv11, introducing nonlinear factors that enable the YOLOv11 network to learn and fit more complex functional relationships. The calculation formula is expressed as follows:(9)SiLU(x)=x⋅Sigmoid(x)(10)$$Sigmoid(x)=\frac{1}{1+e^{-x}}$$

Although the SILU activation function demonstrates excellent performance in terms of nonlinear expression and gradient stability in YOLOv11 [40,41], its capacity to address the vanishing gradient issue is relatively limited, and its stability is significantly influenced by input values and network parameters. To address these challenges, this paper proposes the adoption of the ARelu activation function, which leverages an attention module to learn element-level residuals. For activated inputs, it can effectively amplify the gradients propagated from downstream layers, thereby mitigating the vanishing gradient problem. In the context of cotton leaf disease recognition, the activation effect can be adjusted more flexibly to accommodate complex and multi-scene data. The calculation of ARelu is presented as follows.(11)$$L\left(x_{i},\alpha ,\beta \right)=\left\{\begin{array}{ll} C\left(\alpha \right)x_{i},\;x_{i}<0 & \\ \sigma \left(\beta \right)x_{i},\;x_{i}\geq 0 & \end{array}\right.$$

$C\left(\alpha \right)$,$\sigma \left(\beta \right)$ refers to the “element-wise residual parameters learned by the attention module, adjusting activation intensity adaptively”.

A is the ith element of the input feature map. Combined with Equation (12), the activation function ARelu is obtained, and the corresponding calculation formula is given in Equation (13).(12)$$R\left(x_{i}\right)=\left\{\begin{array}{ll} 0,\;\;x_{i}<0 & \\ x_{i},\;x_{i}\geq 0 & \end{array}\right.$$(13)$$F\left(x_{i},\alpha ,\beta \right)=R\left(x_{i}\right)+L\left(x_{i},\alpha ,\beta \right)=\left\{\begin{array}{ll} C\left(\alpha \right)x_{i},\;{\;\;\;\;\;\;\;\;\;\;\;x}_{i}<0 & \\ \left(1+\sigma \left(\beta \right)\right)x_{i},\;x_{i}\geq 0 & \end{array}\right.$$

$R\left(x_{i}\right)$ is defined as the “base ReLU function, providing basic non-linear transformation”.

$L\left(x_{i},\alpha ,\beta \right)$ is denoted as the “attention-guided residual term, enhancing gradient propagation in deep networks”.

Parameters α and β in the ARelu function are learned adaptively during training to address the variability of cotton leaf disease features: α amplifies gradients for faint early lesions (weak signals) to prevent under-detection, while β modulates responses for intense mature lesions (strong signals) to avoid over-saturation. This adaptive adjustment stabilizes training, especially when diverse lesion morphologies are learned across six disease types.

ARelu is an advanced method that enhances the traditional ReLU activation function by incorporating an attention module for element-level residual learning. This approach addresses the limitations of the traditional ReLU function in handling the vanishing gradient problem. In the conventional ReLU activation function, a positive input directly produces the input value as output, while zero or negative inputs yield an output of zero. Consequently, during forward propagation in multi-layer neural networks, gradients may diminish rapidly, and during backpropagation, they are prone to vanishing entirely. In contrast, ARelu’s design introduces an attention mechanism that enables the effective amplification of gradients propagated from downstream layers through element-level residual learning. As a result, even when the input is zero or negative, ARelu can still learn and preserve critical structural information via the attention mechanism, thereby mitigating the vanishing gradient issue.

#### 2.3.8. Performance Evaluation

For a comprehensive assessment of the model’s efficacy in detecting cotton leaf diseases, we utilized a set of evaluation metrics listed in Table 2, which comprises mAP_0.5, mAP_0.5:0.95, precision, recall, F1 score, and FPS.

TP, FP, FN, and TN denote true positive, false positive, false negative, and true negative, respectively. AP_i represents the mean precision value associated with index i, where M denotes the total number. Among these evaluation criteria, AP represents the average precision across varying recall rates. mAP_0.5 is computed when the intersection-over-union (IoU) threshold is set to 0.5, representing the mean of the AP values at this specific threshold. mAP_0.5:0.95 is a more stringent and comprehensive evaluation metric, which involves calculating the average precision (AP) for each IoU threshold ranging from 0.5 to 0.95, followed by averaging these AP values. Precision is a critical metric for assessing the performance of a classification model, also referred to as the positive predictive value, which quantifies the proportion of true positive predictions among all samples classified as positive by the model. Recall is a critical metric for evaluating the performance of a classification model, representing the proportion of actual positive instances that are correctly identified by the model. The F1 score, defined as the harmonic mean of precision and recall, provides a balanced measure of these two metrics. A higher F1 score indicates superior overall performance of the model in terms of both precision and recall.

#### 2.3.9. Model Environment Configuration

The computer parameter configuration employed by the ACURS-YOLO network for training, testing, and validation is presented in Table 3. The optimizer utilizes Adaptive Moment Estimation (Adam), with the epoch set to 150, the batch size set to 24, and the learning rate configured at 0.001.

## 3. Model Introduction

### 3.1. Performance Comparison of Classical Models

To verify the performance effectiveness of the introduced model, the ACURS-YOLO network was compared with several state-of-the-art object detection models, including YOLOv11, YOLOv10, YOLOv9, YOLOv8, and SSD. The comparison was conducted based on multiple evaluation metrics: mAP_0.5, mAP_0.5:0.95, precision, recall, F1 score, and FPS. Depicted in Table 4 alongside Figure 10, the improved ACURS-YOLO model yielded results of 94.6%, 83.4%, 95.5%, 89.9%, and 92.3% for mAP_0.5, mAP_0.5:0.95, precision, recall, and F1 score, respectively. ACURS-YOLO optimizes the computational complexity via the SimSPPF module. Despite the incorporation of U-Net v2 and the CBAM, the overall frames per second (FPS) value remains at 148, which is only approximately 5% lower than that of YOLOv11. These results indicate that when compared with other models, the presented model not only boosts detection precision but also retains a satisfactory inference rate, thus showing great potential for real-time applications.

As seen in Figure 11, the mAP_0.5 of the six models showed notable variations in the first 130 epochs and steadied gradually over the next 20 epochs. The curve of the ACURS-YOLO network consistently surpassed those of other traditional baseline models shown in the graph, with a final mAP_0.5 of 94.6%. These findings demonstrate that the ACURS-YOLO network exhibits significantly better performance in cotton leaf disease detection compared to other models.

### 3.2. Performance Comparison of Attention Mechanism Modules

In order to evaluate the effectiveness of adding the CBAM attention mechanism module to the ACURS-YOLO network, we refer to the proposed model without CBAM as Model 1; that is, in the ACURS-YOLO network, the added CBAM module is removed, and the model is named Model 1. The proposed ACURS-YOLO network is compared with Model 1, Model 1 + ECA, Model 1 + SE, and YOLOv11. As shown in Figure 12, the model that uses the CBAM module is superior to other models in all evaluation indicators and exhibits a significant improvement compared with the original YOLOv11.

As shown in Table 5 and Figure 13, the enhancements in the performance of ACURS-YOLO are apparent not just in accuracy but also in speed. Compared to Model 1 + SE, ACURS-YOLO maintains nearly the same inference speed while achieving a 4.4% higher mAP_0.5. When compared to Model 1 + ECA, the introduction of CBAM only incurs a 4 FPS reduction, yet it yields significant gains in precision and recall. This validates the finding that the CBAM module effectively enhances feature discrimination with minimal computational overhead. When compared to Model 1 + ECA, the corresponding enhancements are 3.5%, 5.2%, 5.6%, 6.1%, and 5.9%. Furthermore, relative to Model 1 alone, ACURS-YOLO achieves increases of 3.9%, 4.2%, 4%, 1%, and 2.4% in the aforementioned metrics. Through this comparative analysis, it is evident that the CBAM module significantly contributes to the model’s performance, particularly in enhancing detection accuracy, thereby validating its effectiveness.

### 3.3. Activation Function Performance Comparison

To systematically assess the influence of various activation functions on the performance of models for cotton leaf disease detection, this research performs comparative tests utilizing YOLOv11’s default SiLU activation function, the traditional ReLU function, and the improved AReLU function. The ACURS-YOLO network is employed as the baseline model in these experiments. By replacing the activation functions for comparative analysis, this study analyzes core object detection evaluation indicators, including mAP_0.5, mAP_0.5:0.95, precision, recall, and F1 score. Additionally, the convergence characteristics of the mAP_0.5 curve during the training process are discussed. Based on the quantitative findings provided in Table 6 and Figure 14, AReLU not only surpasses SiLU and ReLU in terms of accuracy but also demonstrates reasonable speed trade-offs. AReLU achieves a 2.3% higher mAP_0.5 than SiLU while reducing the inference speed by only 7 FPS. In contrast, although ReLU is the fastest, it shows lower accuracy, highlighting the fact that AReLU effectively mitigates the vanishing gradient problem without sacrificing real-time performance. Specifically, when compared with SiLU, the mAP_0.5 of AReLU increases from 92.3% to 94.6%, while the mAP_0.5:0.95 for AReLU increases from 80.2% to 83.4%. These improvements reflect a comprehensive enhancement in detection accuracy under varying IoU thresholds. In terms of precision, AReLU achieves a 0.3% higher score than SiLU, indicating a reduction in the model’s false detection rate. The improvement in recall is even more pronounced, rising from 81.2% to 89.3%, which suggests that AReLU is more effective at detecting cotton leaf disease targets, thereby minimizing missed detections and holding significant practical implications. The F1 score, as the harmonic mean of precision and recall, rises from 87.6% to 92.3%, showing that the framework attains a superior equilibrium between detection accuracy and completeness. Notably, the FPS indicators in Table 6 correspond to the framework’s real-time capability verified in Table 4. For subsequent ablation studies and scenario testing, the emphasis will be placed on evaluating module synergies and practical generalization capabilities. This is because the core speed–accuracy trade-off of ACURS-YOLO has been systematically verified in prior sections.

The mAP_0.5 training curve in Figure 15 demonstrates that AReLU exhibits a faster convergence rate at the onset of training. As the number of training iterations increases, its performance consistently surpasses that of SiLU and ReLU, eventually stabilizing at a higher level after 130 training epochs. This indicates that AReLU not only enhances the final performance of the model but also improves the stability of the training process. In summary, by optimizing the nonlinear mapping characteristics of the activation function, AReLU significantly improves the detection capability of the ACURS-YOLO network for cotton leaf diseases. It particularly demonstrates substantial advantages in addressing missed detections in complex field environments, thereby providing robust performance support for subsequent model deployment.

### 3.4. Ablation Experiment

To confirm the efficacy of each modified module in the ACURS-YOLO model and their specific contributions to the model’s performance, ablation experiments were performed to assess the performance of different module combinations. In these experiments, the original YOLOv11 was employed as the baseline model (Model 1). Subsequently, the U-Net v2, CBAM, SimSPPF, C3k2_RCM, and ARelu modules were individually incorporated into the model in sequence (Models 2–6). Furthermore, the synergistic effects of multi-module combinations were examined (Models 7–12). Focusing on evaluation indicators, including mAP_0.5, mAP_0.5:0.95, precision, recall, and F1 score, Figure 16 offers a straightforward visualization of the multi-index performance differences among various model configurations for cotton leaf disease detection. The findings reveal that the baseline model attains the lowest performance across all five metrics.

The specific results are presented in Table 7. Based on the research analysis of Models 1 through 6, all improvement components positively enhance model performance. In Model 2, the backbone network is optimized using the concept of medical image segmentation, resulting in an increase in mAP_0.5 from 86.8% to 88.5% and an improvement in mAP_0.5:0.95 by 2.9%, which demonstrates its effectiveness in enhancing multi-scale feature extraction capabilities. In Model 3, channel and spatial attention mechanisms are introduced, leading to increases in precision and recall of 5.2% and 3.9%, respectively, thereby validating the attention mechanism’s ability to focus on key disease features. Model 4 enhances computational efficiency while maintaining accuracy via parallel pooling and ReLU activation optimization, with mAP_0.5 increasing from 86.8% to 88.2% (a 1.4% improvement). Model 5 improves neck feature fusion, resulting in a 5.9% increase in recall, indicating stronger detection capabilities for multi-scale targets in complex scenarios. Finally, Model 6 mitigates the vanishing gradient problem through adaptive residual activation, achieving a 2.9% increase in the F1 score, which reflects improvements in both model training stability and detection integrity.

According to the analysis of Models 7–12, the performance of Models 7–11 surpasses that of the baseline model, demonstrating a significant synergistic effect. Specifically, when ARelu is absent in Model 7, its mAP_0.5 reaches 92.3%, which is marginally lower than Model 12 but remains higher than most comparative models. Model 8 lacks the integration of C3k2_RCM, resulting in a drop in recall to 85% and thus underscoring the significance of the neck module in modeling contextual information. In Model 11, the absence of U-Net v2 causes the mAP_0.5:0.95 to drop to 78.9%, underscoring the critical role of backbone network enhancements for fundamental feature extraction. Ultimately, Model 12 integrates all modules, achieving an mAP_0.5 of 94.6% and an mAP_0.5:0.95 of 83.4%. Additionally, the precision rate, recall rate, and F1 score reach 95.5%, 89.3%, and 92.3%, respectively, showing an overall improvement relative to the baseline framework. Such findings demonstrate that each module collectively contributes to boosting the detection capability for cotton leaf diseases in intricate environments through structural optimization and functional complementarity.

Ablation study findings reveal that U-Net v2’s feature extraction capacity, CBAM’s attention mechanism, SimSPPF’s effective multi-scale fusion, C3k2_RCM’s context modeling, and ARelu’s activation optimization all play a significant role in enhancing the model’s performance. Furthermore, the synergistic effect between these modules is considerable. These findings validate the scientific rationale and necessity of the ACURS-YOLO model design.

### 3.5. Model Testing

To further validate the effectiveness of the presented framework, the specimens were evaluated using both the ACURS-YOLO and YOLOv11 networks. In the test set, 20 cotton leaf disease images with complex background environments were randomly selected for each category, resulting in a total of 120 images used to assess the performance of the ACURS-YOLO network. The original model, YOLOv11, was chosen as a benchmark for comparison. Detailed findings are shown in Table 8, where these values represent the counts of accurately recognized images.

Through comparative tests in complex scenes, as shown in Figure 17, the ACURS-YOLO model shows an excellent generalization effect. Although there are still a small number of misjudgments in extreme occlusion and highly similar disease scenes, the overall performance of ACURS-YOLO is significantly better than that of the original YOLOv11. When YOLOv11 detects a variety of cotton leaf diseases under actual conditions, there are some omissions, and the recognition accuracy is low. The ACURS-YOLO network enhances the detection ability in complex scenes and realizes more accurate leaf recognition.

Through the abovementioned experimental analyses, it becomes clear that the UNetv2 image segmentation model, which was initially developed for medical scenarios, is also suitable for cotton leaf disease identification and shows certain improvements compared to the original YOLOv11 framework. In comparison with the baseline model, the ACURS-YOLO network exhibits substantial improvements in detecting six types of cotton leaf diseases. As shown in Table 8, the average accuracy of the ACURS-YOLO network in detecting cotton leaf diseases exceeds that of the original YOLOv11 by 19.2%. Figure 18 displays the confusion matrix regarding the detection of six types of cotton leaf diseases. It can be seen that the YOLOv11 model incurs certain errors in environments with complex backgrounds, whereas the ACURS-YOLO network achieves a notable reduction in recognition errors relative to YOLOv11. However, there remains a risk of misclassification. Based on the above experimental findings, the improved model shows significant advancements in the identification capability of cotton leaf diseases.

## 4. Discussion

In this research, we successfully developed an advanced cotton leaf disease identification model designated as ACURS-YOLO, which was built upon the improved YOLOv11 architecture. Experimental findings show that the model delivers superior performance. In comparison with classical models, ACURS-YOLO markedly surpasses the original YOLOv11 and other state-of-the-art object detection models with respect to mAP_0.5, mAP_0.5:0.95, precision, recall, and F1 score. This indicates that the improvement strategies implemented in the model are highly effective. From the perspective of the improved modules, the U-Netv2 module was integrated into the backbone network of YOLOv11 to enhance the model’s multi-scale feature extraction capability. The successful experience gained from medical image segmentation has proven to be applicable to cotton leaf disease detection, effectively improving the model’s ability to capture complex disease features. The integration of the CBAM attention mechanism allows the model to focus more effectively on critical disease features, thereby significantly improving detection accuracy. By utilizing both channel and spatial attention modules, the model achieves a more precise emphasis on disease-related regions. The SimSPPF module optimizes multi-scale feature fusion, thereby enhancing computational efficiency while maintaining accuracy and addressing the high computational latency and complexity issues inherent in the traditional SPPF module. The C3k2_RCM module enhances neck feature fusion, boosting the model’s capacity to detect multi-scale targets in complex environments and offsetting the contextual modeling constraints of the original C3k2 module. The ARelu activation function alleviates the gradient vanishing problem, thereby optimizing model training stability and detection integrity, and effectively reduces false negatives in challenging field environments [42,43].

Compared with prior studies, this study innovatively incorporated a variety of improvement strategies to tackle the complexity in cotton leaf disease detection, thereby addressing issues related to the high similarity and large detection errors associated with certain diseases inherent in traditional detection methods. Nevertheless, this study has certain limitations. Although the model’s recognition capability for complex backgrounds has been improved, there is still a risk of misclassification. Furthermore, model performance may be affected in extremely challenging environments, such as severely damaged leaves, significant occlusion, or extremely poor lighting conditions.

Future research may proceed as follows: First, the model architecture can be further optimized by investigating more effective feature extraction and fusion methods. For example, advanced attention mechanisms or neural network structures might be integrated to boost the model’s robustness in complex environments. Second, the dataset’s scale and diversity should be expanded to encompass images of leaf diseases from various growing conditions and cotton varieties, thus enhancing the model’s generalizability. Third, efforts could be made to deploy the model on mobile terminals while integrating additional sensor data, such as spectral and multimodal data, to enrich the information available for disease detection and thus improve the accuracy and reliability of the system.

## 5. Conclusions

Aiming to address the critical challenges in cotton leaf disease detection, such as complex background interference, missed detection of small target lesions, and insufficient generalization for multi-form diseases, this study proposes an ACURS-YOLO detection network that integrates medical image segmentation concepts with lightweight improvement strategies. This provides an efficient solution for automated disease monitoring in smart sensors. Built upon YOLOv11, the proposed network incorporates multi-scale feature enhancement, an adaptive attention mechanism, and a robust training strategy through cross-domain knowledge transfer and multi-module collaborative optimization. These enhancements significantly improve the model’s detection performance in real-world field environments.

Firstly, we developed a specialized dataset comprising six typical cotton leaf diseases (Cotton Leaf Spot, Cotton Leaf Curl, Cotton Brown Spot, Cotton White Mold, Cotton Verticillium Wilt, and Cotton Fusarium Wilt). Through field collection and data augmentation techniques, the dataset was expanded to include 3000 samples, encompassing various lighting conditions, degrees of leaf occlusion, and stages of disease progression. This dataset provides robust and reliable support for model training and validation. To address the limitations of traditional YOLO series models in small target detection and adaptability to complex backgrounds, this study innovatively incorporates the U-Net v2 module, which is commonly used in medical image segmentation, to reconstruct the backbone network. By utilizing its encoder–decoder framework, the model strengthens its multi-scale feature fusion ability, allowing for the effective capture of fine-grained textures and contextual associations of leaf disease lesions.

Meanwhile, through the integration of the CBAM attention module, the model can dynamically focus on disease regions, reduce interference from leaf texture and environmental noise, and boost the precision of feature selection. In the neck network architecture, the C3k2_RCM module is integrated to reinforce the modeling of long-range contextual dependencies via a rectangular self-calibration mechanism, thereby addressing the issue of the insufficient exploration of multi-scale target semantic correlations in the original model. The enhanced SimSPPF module employs a parallel pooling structure to reduce computational complexity while preserving multi-scale feature extraction capabilities, thus improving inference speed. Additionally, the ARelu activation function alleviates the vanishing gradient problem through adaptive residual learning and improves the training stability of the deep network model.

The experimental results demonstrate that ACURS-YOLO significantly outperforms the original YOLOv11 and classical object detection models, such as SSD and other established models, in key performance indicators. From a practical application standpoint, ACURS-YOLO exhibits remarkable generalization capabilities when evaluated in complex scenarios. Notably, the recognition accuracy for easily confused diseases has improved by 19.2% compared to the original model, while ACURS-YOLO maintains a rate of 148 FPS—close to YOLOv11’s 156 FPS—demonstrating its practical value in real-time agricultural monitoring. This advancement effectively addresses the issue of misclassification caused by high disease similarity in traditional approaches.

However, this study has certain limitations that warrant attention. While the model demonstrates strong performance in conventional field environments, its detection accuracy tends to decrease under extreme conditions, such as severe leaf occlusion, extremely low light levels, or the early miniaturization of disease spots. Furthermore, although the dataset encompasses six primary diseases, it lacks representation of disease characteristics specific to different cotton varieties (e.g., insect-resistant cotton and long-staple cotton), which could potentially compromise the model’s generalizability across diverse planting scenarios. Going forward, with additional polishing and tuning, the ACURS-YOLO network model is poised for wider use in agricultural image detection, thus promoting a shift toward enhanced intelligence and precision in agricultural production.

## Figures and Tables

**Figure 1 sensors-25-04432-f001:**
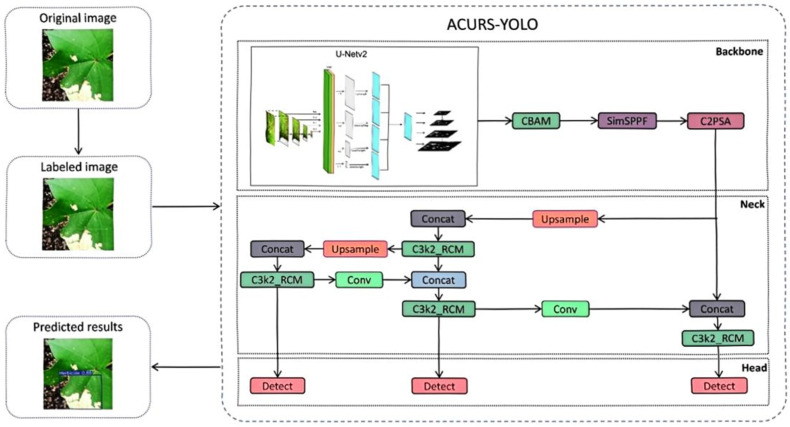
Implementation process of cotton disease identification.

**Figure 2 sensors-25-04432-f002:**
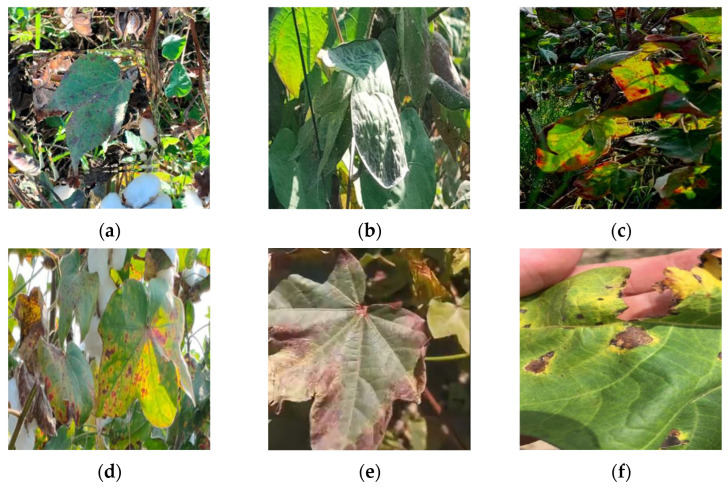
Some images sourced from the dataset: (**a**) Cotton Leaf Spot, (**b**) Cotton Leaf Curl, (**c**) Cotton Brown Spot, (**d**) Cotton White Mold, (**e**) Cotton Verticillium Wilt, (**f**) Cotton Fusarium Wilt.

**Figure 3 sensors-25-04432-f003:**
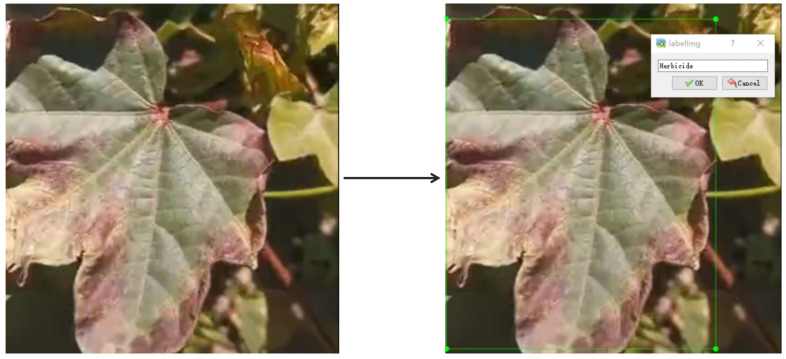
LabelImg was used to label cotton leaf disease images.

**Figure 4 sensors-25-04432-f004:**
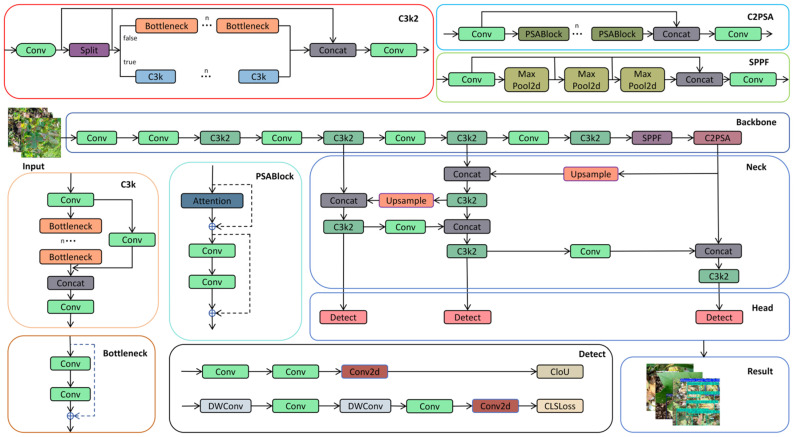
Overall structure of YOLOv11.

**Figure 5 sensors-25-04432-f005:**
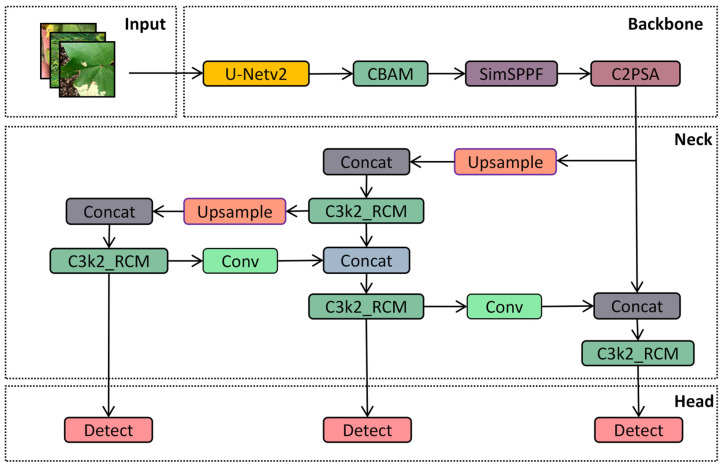
Diagram of the structure of ACURS-YOLO.

**Figure 6 sensors-25-04432-f006:**
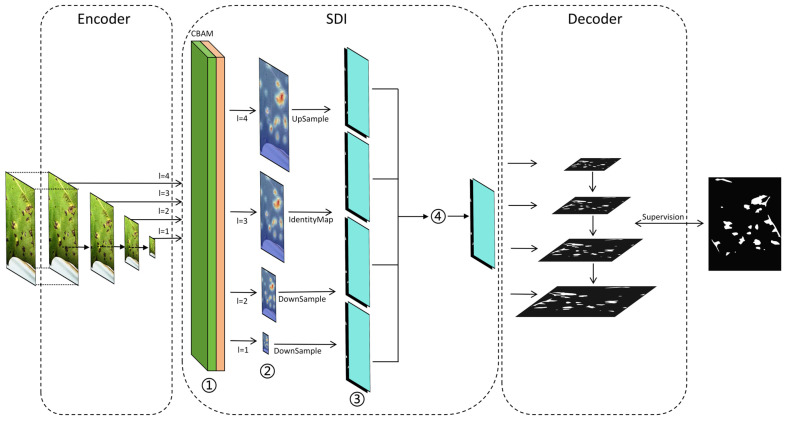
Overall structural diagram of U-Net v2. In the figure, ①, ②, ③, and ④ correspond respectively to (1), (2), (3), and (4) as shown in the notice.

**Figure 7 sensors-25-04432-f007:**
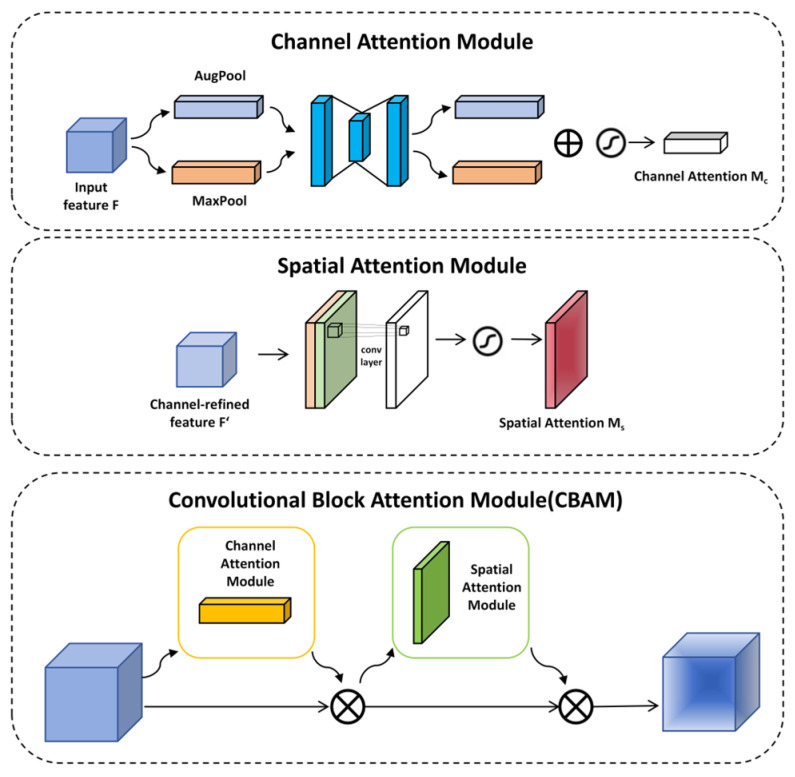
Overall structural diagram of the CBAM.

**Figure 8 sensors-25-04432-f008:**
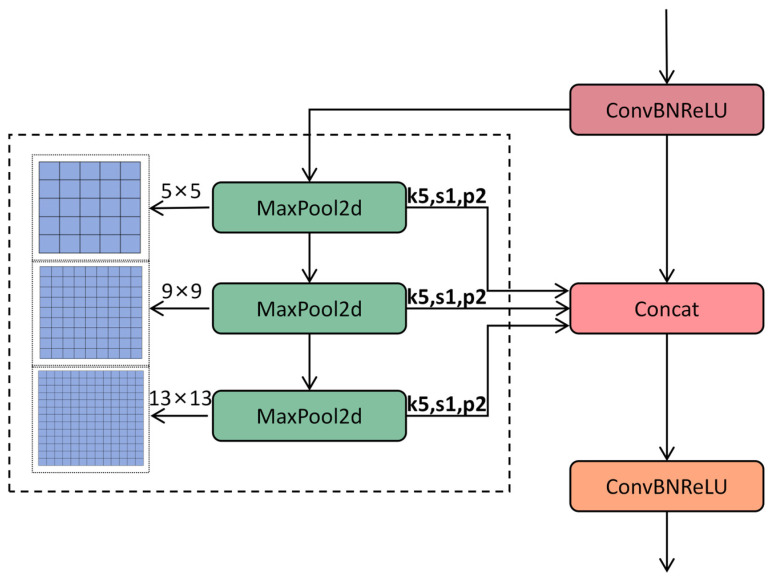
Overall structural diagram of SimSPPF.

**Figure 9 sensors-25-04432-f009:**
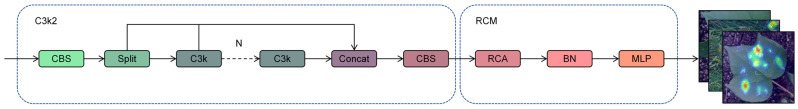
Overall structural diagram of C3k2_RCM.

**Figure 10 sensors-25-04432-f010:**
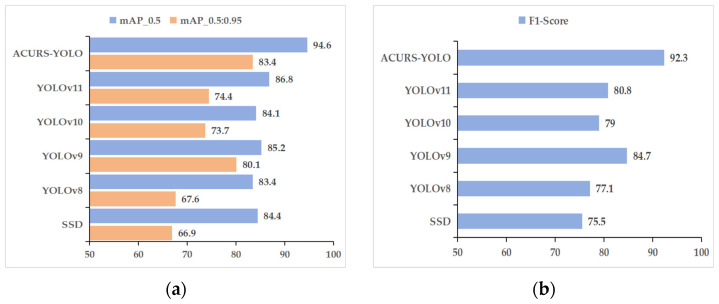
Performance comparison of classical models. (**a**) mAP_0.5 and mAP_0.5:0.95, (**b**) F1 score.

**Figure 11 sensors-25-04432-f011:**
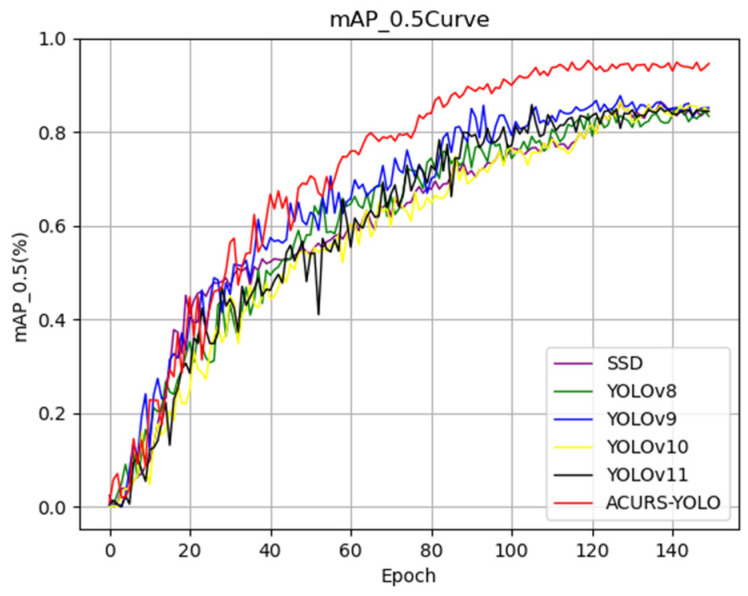
The classical model’s mAP_0.5 curve.

**Figure 12 sensors-25-04432-f012:**
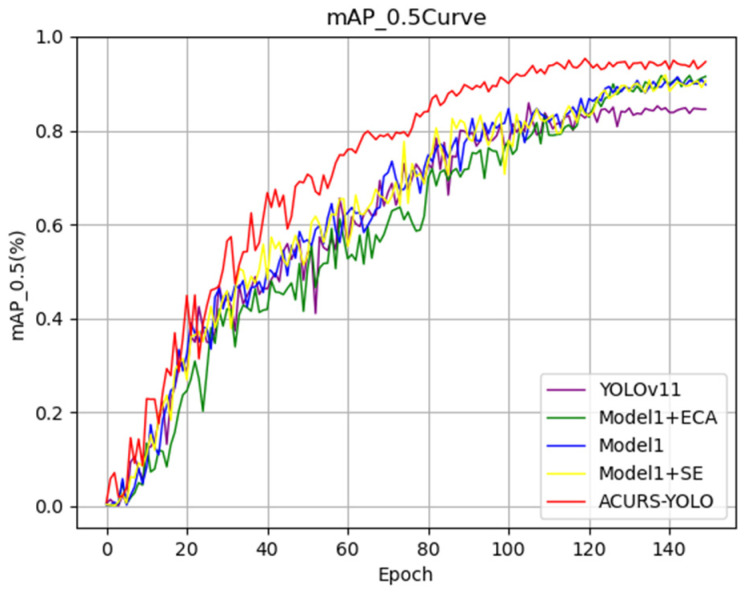
Performance comparison of the attention mechanism modules’ mAP_0.5 curves.

**Figure 13 sensors-25-04432-f013:**
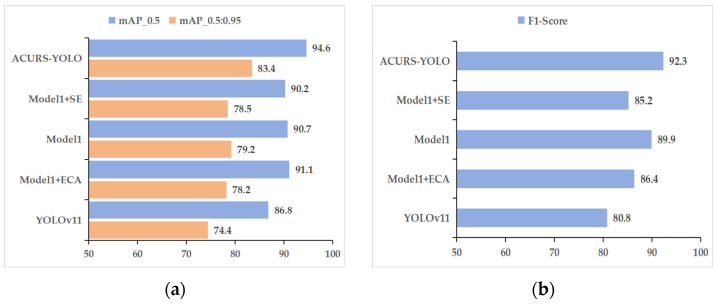
Efficacy contrast of attention mechanism components. (**a**) mAP_0.5 and mAP_0.5:0.95, (**b**) F1 score.

**Figure 14 sensors-25-04432-f014:**
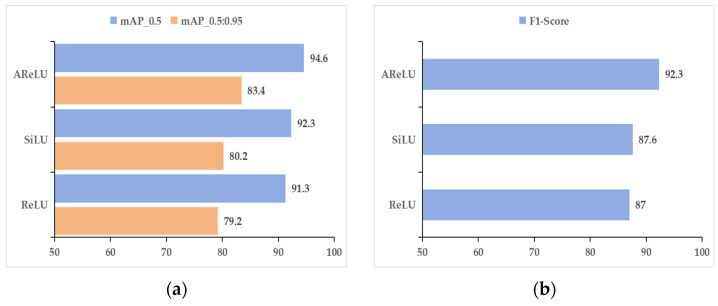
Activation function performance comparison: (**a**) mAP_0.5 and mAP_0.5:0.95, (**b**) F1 score.

**Figure 15 sensors-25-04432-f015:**
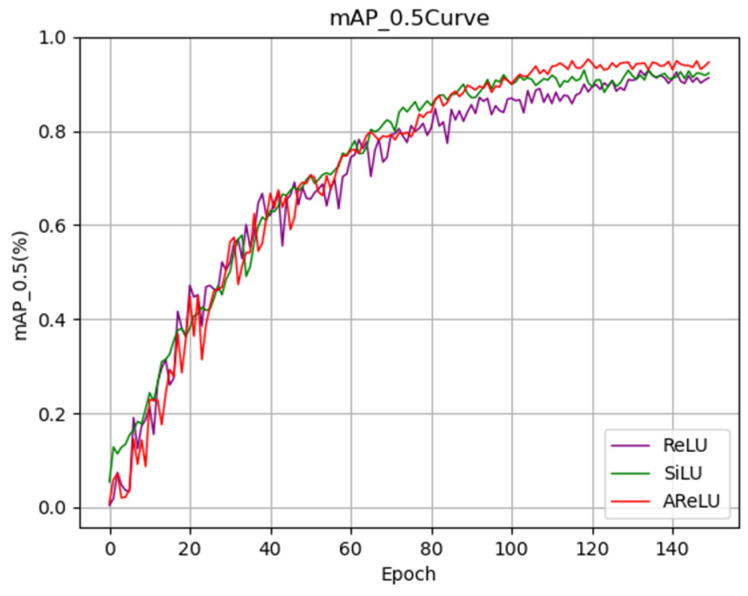
Activation function performance comparison of the mAP_0.5 curve.

**Figure 16 sensors-25-04432-f016:**
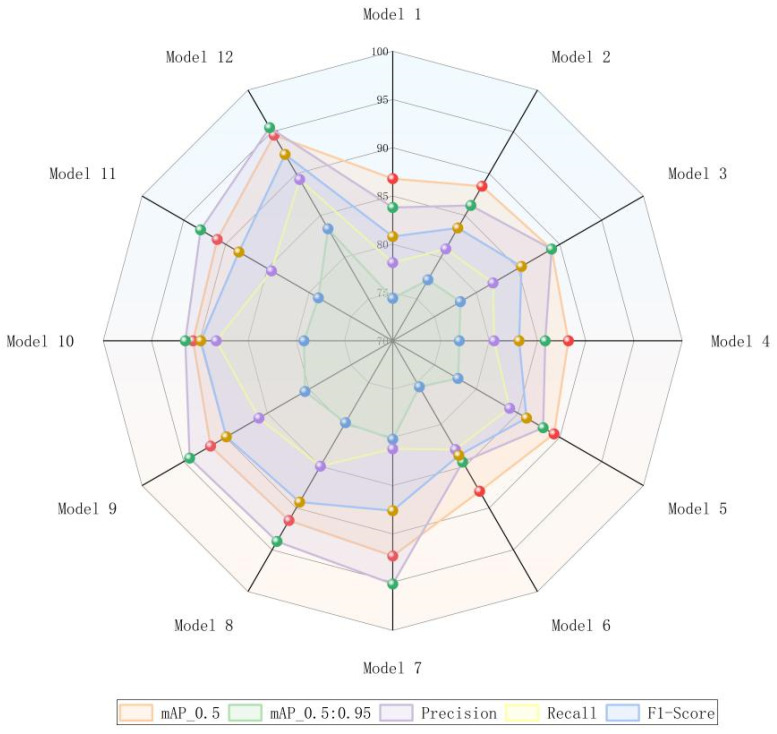
Multi-metric comparison graph demonstrating the influence of each enhanced module on the efficacy of the cotton leaf disease detection framework in the fusion experiment.

**Figure 17 sensors-25-04432-f017:**
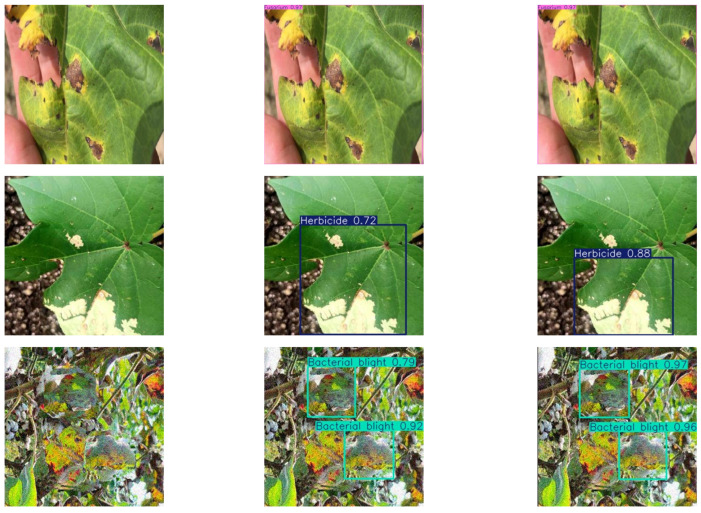
Model test comparison images: (**a**) original image, (**b**) YOLOv11, (**c**) ACURS-YOLO.

**Figure 18 sensors-25-04432-f018:**
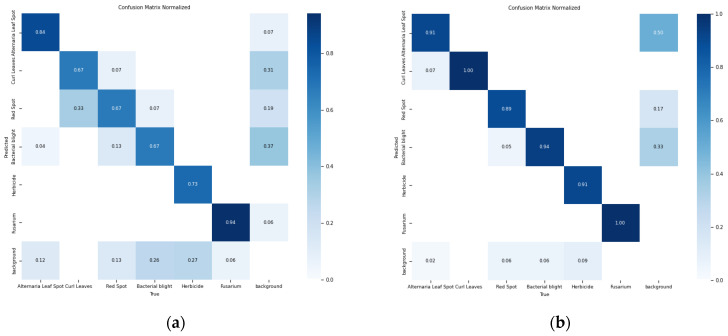
Confusion matrix for leaf disease detection in six cotton species: (**a**) YOLOv11, (**b**) ACURS-YOLO.

**Table 1 sensors-25-04432-t001:** Cotton leaf disease dataset.

Diseases	Training Set	Validation Set	Test Set	Total
Cotton Leaf Spot	400	50	50	500
Cotton Leaf Curl	400	50	50	500
Cotton Brown Spot	400	50	50	500
Cotton White Mold	400	50	50	500
Cotton Verticillium Wilt	400	50	50	500
Cotton Fusarium Wilt	400	50	50	500

**Table 2 sensors-25-04432-t002:** Performance evaluation metrics.

Index	Equation
mAP	$\frac{\sum_{i=1}^{M}{AP}_{i}}{M}$
Precision	$\frac{TP}{TP+FP}$
Recall	$\frac{TP}{TP+FN}$
F1 Score	$\frac{2\times Precision\times Recall}{Precision+Recall}$

**Table 3 sensors-25-04432-t003:** Computer parameter configuration.

Configuration	Parameter
Operating System	Windows10 Professional
CPU	Intel(R) Xeon(R) Silver 4310 CPU @ 2.10 GHz
GPU	NVIDIA GeForce RTX 4090
Memory	127 GB
Language	Python 3.8
Cuda	12.4
Framework	Pytorch 1.8.1

**Table 4 sensors-25-04432-t004:** Traditional model performance contrast metrics.

Model	mAP_0.5	mAP_0.5:0.95	Precision	Recall	F1 Score	FPS
SSD	84.4	66.9	85.9	67.3	75.5	52
YOLOv8	83.4	67.6	74.9	79.5	77.1	110
YOLOv9	85.2	80.1	87.2	82.3	84.7	135
YOLOv10	84.1	73.7	87.1	72.3	79.0	142
YOLOv11	86.8	74.4	83.8	78.1	80.8	156
ACURS-YOLO	94.6	83.4	95.5	89.3	92.3	148

**Table 5 sensors-25-04432-t005:** Classical model performance comparison values.

Model	mAP_0.5	mAP_0.5:0.95	Precision	Recall	F1 Score	FPS
YOLOv11	86.8	74.4	83.8	78.1	80.8	156
Model 1 + ECA	91.1	78.2	89.9	83.2	86.4	152
Model 1	90.7	79.2	91.5	88.3	89.9	154
Model 1 + SE	90.2	78.5	91.8	79.5	85.2	149
ACURS-YOLO	94.6	83.4	95.5	89.3	92.3	148

**Table 6 sensors-25-04432-t006:** Activation function performance comparison values.

Model	mAP_0.5	mAP_0.5:0.95	Precision	Recall	F1 Score	FPS
ReLU	91.3	79.2	94.4	80.7	87.0	158
SiLU	92.3	80.2	95.2	81.2	87.6	155
AReLU	94.6	83.4	95.5	89.3	92.3	148

**Table 7 sensors-25-04432-t007:** Results of ablation experiments.

Number	U-Netv2	CBAM	SimSPPF	C3k2_RCM	ARelu	mAP_0.5 (%)	mAP_0.5:0.95 (%)	Precision (%)	Recall (%)	F1 Score (%)
Model 1						86.8	74.4	83.8	78.1	80.8
Model 2	√					88.5	77.3	86.2	81	83.5
Model 3		√				89	78.1	89	82	85.4
Model 4			√			88.2	76.9	85.8	80.5	83.1
Model 5				√		89.3	77.8	88	84	86
Model 6					√	88	75.5	84.5	83	83.7
Model 7	√	√	√	√		92.3	80.2	95.2	81.2	87.6
Model 8	√	√	√		√	91.5	79.8	94	85	89.3
Model 9	√	√		√	√	91.8	80.5	94.3	86	89.9
Model 10	√		√	√	√	90.7	79.2	91.5	88.3	89.9
Model 11		√	√	√	√	91	78.9	93	84.5	88.4
Model 12	√	√	√	√	√	94.6	83.4	95.5	89.3	92.3

**Table 8 sensors-25-04432-t008:** ACURS-YOLO network test results.

Case	ACURS-YOLO	YOLOv11
Cotton Leaf Spot	18	17
Cotton Leaf Curl	20	13
Cotton Brown Spot	18	13
Cotton White Mold	19	13
Cotton Verticillium Wilt	18	15
Cotton Fusarium Wilt	20	19
Average precision	94.2%	75.0%

## Data Availability

The data involved in the study can be obtained by contacting the authors.

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
