# Peer review of "Investigation of an Efficient Multi-Class Cotton Leaf Disease Detection Algorithm That Leverages YOLOv11"

_sensors, 2025, doi:10.3390/s25144432_

Round 1
Reviewer 1 Report
Comments and Suggestions for Authors
The ACURS-YOLO model proposed in this manuscript is improved in various aspects based on YOLOv11 by integrating U-Net v2, CBAM attention mechanism, SimSPPF structure, C3k2_RCM module with ARelu activation function. The focus of the manuscript is to improve the real-time detection of multiple types of cotton leaf diseases, and a systematic evaluation is carried out on the basis of constructing a dataset. The research has certain application value, especially in the agricultural intelligent detection scenario with the potential to be promoted. However, the manuscript still has the following problems, which need to be further improved.
Section 2.1:
“Experimental design”, it is mentioned that a dataset containing six representative foliar diseases of cotton was constructed (leaf spot, leaf rolling, brown spot, white mold, verticillium wilt, fusarium wilt). However, the names Alternaria Leaf Spot, Curl Leaves, Red Spot, Bacterial Blight, Herbicide, and Fusarium appear in the image descriptions in Figure 2. Please be sure to standardize the names of all diseases in your paper to avoid confusion.
Section 2.2:
Is there an imbalance in the dataset in terms of sampling of disease categories under different background conditions (light/leaf shading/early stage of lesions)? How can the impact of this bias on the generalization ability of the model be controlled? Meanwhile, for publicly available datasets, please provide specific citations or dataset names.
The dataset covers six common diseases. In a real cotton field, multiple diseases or stresses (e.g., disease and insect pests, malnutrition symptoms) may co-occur on the same leaf or exhibit visually similar symptoms. How does your model handle the coexistence of multiple diseases? And how well does the model recognize disease types or non-disease stresses outside of the training data?
Section 3.1:
In the manuscript, the expression mAP_0.05 appears several times, please verify whether it should be mAP_0.5. In addition, regarding the nomenclature of mAP_0.5, at present there are various forms such as map_0.5, MAP_0.5 and mAP_0.5, etc., and it is suggested to standardize it into a standard nomenclature.
Table 4 shows that YOLOv9 outperforms YOLO11 in all metrics except mAP_0.5, which is slightly lower than YOLO11, which is the baseline for this manuscript. Does this imply that some of the designs of YOLOv11 are degraded in agricultural scenarios? Why is YOLOv11 still chosen as the base framework instead of the better performing YOLOv9?
Section 3.5: The manuscript mentions that “the ACURS-YOLO network correctly recognized 19.2% more cotton leaf diseases than the original YOLOv11 model” and cites Table 7. However, Table 7 mainly shows the results of the ablation experiments (mAP, Precision, Recall, F1-Score), while Table 8 shows the number of correctly recognized images and the average precision in the model test. According to the data in Table 8, the average accuracy of ACURS-YOLO is 94.2% and that of YOLOv11 is 75.0%, which is a difference of 19.2%. Please make sure that the “correct recognition rate” here refers to the average accuracy and that the quoted table is accurate.
Reviewer 2 Report
Comments and Suggestions for Authors
This study aims to develop ACURS-YOLO, an enhanced YOLOv11 framework integrating segmentation and attention modules to achieve fast, accurate, and robust cotton leaf disease detection in complex field conditions.
- The manuscript exhibits a high similarity index (over 19%), which must be reduced to below 10% to meet publication standards. Also, add the workflow of the other sections at the end of Introduction section.
- The paragraphs in the paper are overly long and difficult to follow; please break them into shorter, more manageable sections.
- In what ways do the integrated modules, U-Net v2, SimSPPF, C3k2_RCM, and ARelu, address complex backgrounds, small-spot detection, and training stability in the ACURS-YOLO architecture? How does ACURS-YOLO compare to baseline YOLO models in terms of disease detection accuracy, inference speed, and overall performance on cotton leaf disease scenarios? In what ways do the U-Net v2, CBAM, SimSPPF, C3k2_RCM, and ARelu modules each contribute to improving feature extraction, attention focus, computational efficiency, and training stability?
- All variables used in the equations should be properly defined. Additionally, please explain how each variable was chosen to obtain the reported results.
- Based on Figure 16, which individual module yields the largest increase in mAP₀.₅ compared to the baseline, and how do combinations of modules (models 7–12) further boost the overall detection performance across the five metrics?
- How does ACURS-YOLO’s generalization performance in complex scenes (Figure 17) compare to the original YOLOv11? Could you add another example like figure 17? What kinds of misjudgments does ACURS-YOLO still encounter in scenarios with extreme occlusion or highly similar diseases? In real-world tests, what omissions and accuracy issues are observed with the original YOLOv11 when detecting various cotton leaf diseases?
Round 2
Reviewer 1 Report
Comments and Suggestions for Authors
Thanks for authors's thorough response and revision and my concerns have been addressed.
Reviewer 2 Report
Comments and Suggestions for Authors
The authors have made the necessary changes in the revised version of the manuscript.